# An Accurate Switching Current Measurement Based on Resistive Shunt Applied to Short Circuit GaN HEMT Characterization

**Carmine Abbate [1], Leandro Colella [1], Roberto Di Folco [1], Giovanni Busatto [2], Emanuele Martano [2], Simone Palazzo [2], Annunziata Sanseverino [2,*] and Francesco Velardi [2]**

[1] DAC Engineering & Reserch S.r.l., Via S. Giovanni Battista 2, 03037 Pontecorvo, Italy; carmine.abbate@daceng.it (C.A.); leandro.colella@daceng.it (L.C.); roberto.difolco@daceng.it (R.D.F.)

[2] DIEI Department of Electrical and Information Engineering, University of Cassino and Southern Lazio, Via G. Di Biasio 43, 03043 Cassino, Italy; busatto@unicas.it (G.B.); emanuele.martano@outlook.com (E.M.); simone.palazzo@unicas.it (S.P.); velardi@unicas.it (F.V.)

[*] Correspondence: a.sanseverino@unicas.it

**Abstract:** The use of a resistive shunt is one of the simplest and most used methods for measuring current in an electronic device. Many researchers use this method to measure drain current during short-circuiting of fast devices such as GaN HEMTs. However, the high switching speed of these devices together with the non-ideality of the shunt resistors produces an overestimation of the current in the initial phases of the transient. In this paper, a passive compensation network is proposed, which is formed by adding an inductor to the voltage measurement circuit and allows an accurate measurement of the current using the resistive shunt even in the presence of very fast devices. The proposed method is validated by simulations and experimental measurements.

**Keywords:** short circuit tests; GaN HEMT; switching current measurement

## 1. Introduction

GaN devices have become more than just a promise to designers. Today, GaN HEMTs represent a mature technology widely used in aerospace [1], automotive industry [2], electric vehicles [3] and for the realisation of power converters where are superior to conventional silicon (Si)-based devices in terms of switching frequency, power rating, thermal capability and efficiency [4].

The use of GaN transistors supports key RF demands such as high gain, low power consumption, high throughput and extremely fast switching speeds. The high breakdown voltage and high power density allow one to operate at higher voltages and manufacture smaller devices while reducing the overall dimensions of power converters. Their small size, together with the high carrier saturation velocity in GaN, allow operations at higher switching frequencies than MOSFETs without incurring high switching losses as the parasitic capacitances of GaN devices are significantly lower than those of MOSFETs, as well as they exhibit reduced conduction losses [5].

However, the high speed of these devices requires great engineering skills both for the development of high-speed driver circuits and in the testing of the switching characteristics. From this point of view, a crucial point is the measurement of the current waveform. The knowledge of this quantity is very important to validate the models of the power devices and to calculate the efficiency of the converter. Moreover, it allows monitoring of circuit operation and triggering protection circuits in case of overcurrent or short circuit. Since GaN devices have very fast transients, of the order 10 A/ns, the current probes must have a very high bandwidth and small insertion impedances to avoid large insertion losses and large effects of parasitic elements on the measurement accuracy.

The Hall effect current probe, Rogowski coil and current shunt are usually used to measure current in power converters. More recently high-bandwidth integrated probe developed for fast GaN devices has been presented [6]. The resistive shunt is by far the simplest sensor to make, cheap and easy to integrate. It can be operated over a wide bandwidth and allows the measurement of both DC and AC components [7]. For this reason, it is widely used in applications such as the characterisation of the behaviour of GaN devices during the short circuit where it is necessary to monitor the current during the entire transient [8–13].

Nevertheless, the insertion of a resistive shunt in a circuit involves the presence of a parasitic inductance which may play a major role in the measurement if the current has very fast transients. In fact, since the current is measured indirectly as the voltage across the impedance, the presence of a very fast gradient induces an overvoltage $\Delta V = L_s \, \Delta I / \Delta t$ on the parasitic inductance that can be mistaken for an overcurrent on the device under test (DUT).

In this paper, we show how it is possible to eliminate measurement artefacts through the insertion of a compensation network which is formed by adding an inductor to the voltage measurement circuit.

## 2. Simulations Results

In ref. [9] we have presented the circuit of Figure 1 used to perform an analysis on the failure mechanisms of GaN power HEMT in short circuit operations.

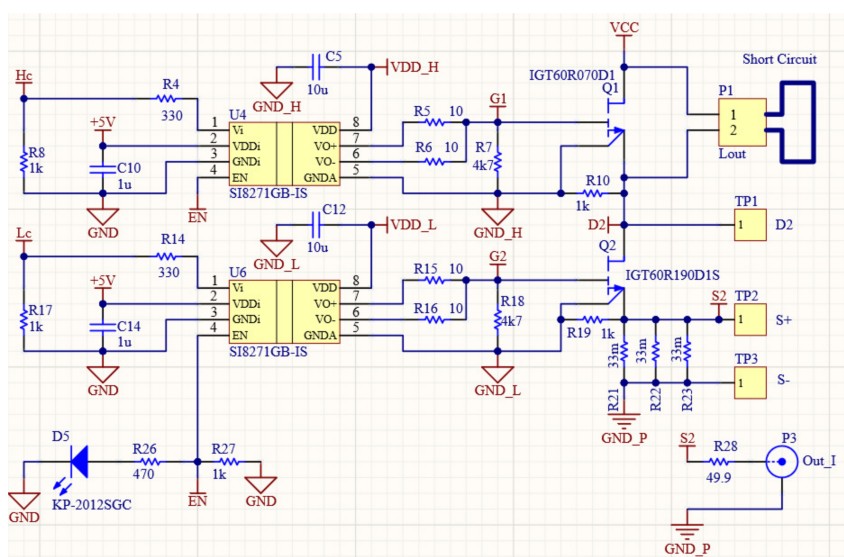

**Figure 1.** Test circuit used for the short circuit measurements.

The circuit consists of two GaN devices (Q1, Q2) connected in a single-leg inverter configuration. The two isolated drivers (U4, U6) allow separate switching of the two devices through the Hc and Lc signals generated by an FPGA-based control circuit. The printed circuit board was designed to minimise parasitic inductance in the power stage and gate branches to minimise drain overvoltages and gate oscillations triggered by very fast transients. The stray inductance of the drain path measured with a vector network analyser was 8 ns. With the same instrument, we also measured the stray inductance of resistive shunt that was about 3 nH. The output current was measured as the voltage drop across a shunt resistance RSH = 11 mΩ obtained as the parallel of three 33 mΩ resistors. This voltage is fed to the input of oscilloscope through the resistor R28 = 50 Ω, used to match the output impedance of the circuit with the input impedance of the cable and oscilloscope.

Figure 2 shows the typical waveforms of gate voltage and drain currents obtained during a 10 µs short circuit test. The tested device was a commercially available 7.5 A–650 V GaN

Power HEMTs. The test conditions refer to two voltage values $V_{CC}$ = 50 V and 200 V and $V_{GS}$ = 6 V.

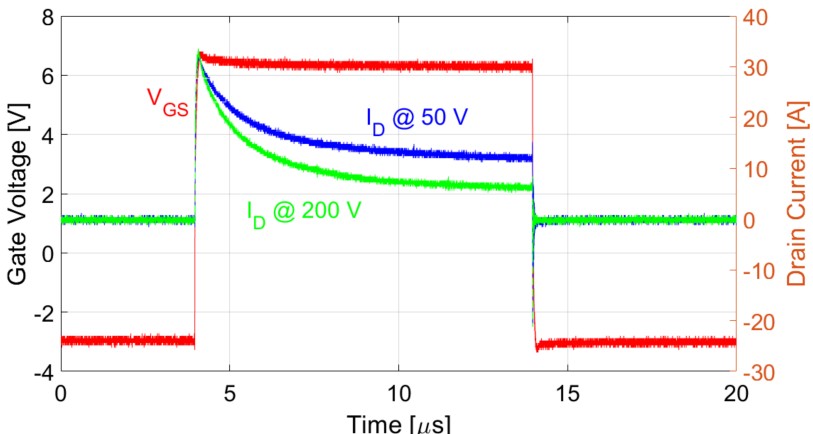

**Figure 2.** Gate voltage (red) and drain currents were measured during the short circuit test at $V_{CC}$ = 200 V (green) and $V_{CC}$ = 50 V (blue).

The transient starts at about 4 us when the gate voltage $V_{GS}$ is raised to 6 V. The current increases up to about 30 A and subsequently decreases due to the occurrence of thermal phenomena related to strong negative temperature feedback of the DUT conductivity [8,12]. However, the trend of the current during its fast variations can be largely altered when using the resistive shunt measurement technique due to the stray inductance which derives from the non-ideality of the resistors.

This issue can be analysed by using PSpice simulator. Using the simulator allows us to evaluate the effective current in the device and to highlight the effect of the parasitic parameters on the measurement.

Figure 3 shows the simulated circuit to reproduces the pulsed tests. The circuit presents two devices in series each with their own driver circuit. The devices used are two 650 V enhancement-mode GaN transistors marketed by GaNSystem. Spice models supplied by the manufacturer were used for the simulations without taking thermal effects into account. Finally, the parallel of the shunt resistors used for measuring the current has been replaced by a single resistor with a series stray inductance of 3 nH. The two 50 Ω resistors account for the impedance matching between the test circuit and the oscilloscope.

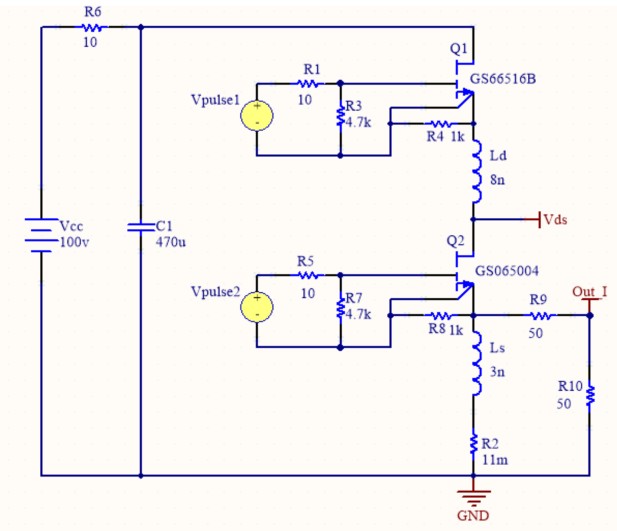

**Figure 3.** Test circuit to observe the pulse characteristics.

The circuit was used to simulate a 1-µs short circuit test. In Figure 4 the waveforms of the drain voltage and drain current obtained with $V_{DS}$ = 100 V and $V_{GS}$ = 6 V are reported. In blue we report the actual current, $I_d$, circulating in the device. In green, the current $I_{sh}$ which would result from the measure of the voltage across the shunt resistor considering the attenuation factor is reported. As can be seen, due to the stray inductance of the resistors, the current $I_{sh}$ presents an overshoot lasting a hundred nanoseconds during which it reaches a value of almost an order of magnitude higher than the steady-state value.

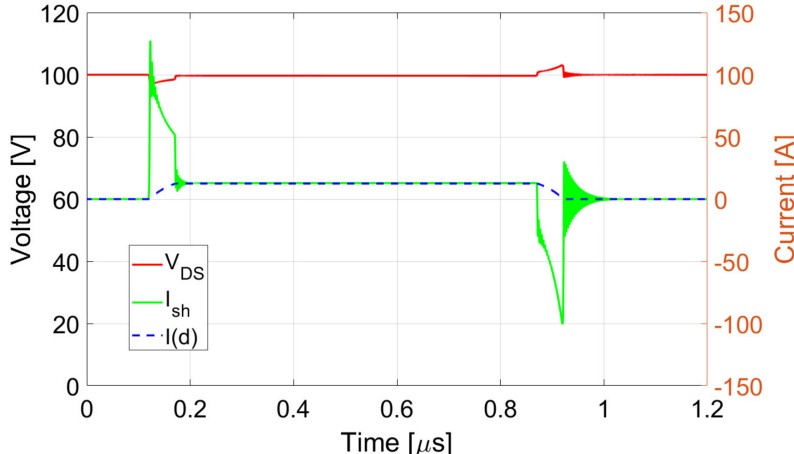

**Figure 4.** Simulated drain voltage and drain current waveforms measured during the short circuit test at $V_{DS}$ = 100 V and $V_{GS}$ = 6 V. I(d) represents the actual current, $I_{sh}$ the current obtained by using the resistive shunt.

Since we have not considered any thermal effect, this behaviour is to be attributed exclusively to the presence of parasitic elements on the measurement circuit. The effect of these parasitic elements can be compensated by introducing an additional inductance, Ls1, in the voltage measurement circuit as shown in Figure 5. The proposed network is the dual method that commonly is applied on attenuated voltage probes of the oscilloscope. In this case, a capacitor in parallel to the voltage divider compensates for the input capacitance of the oscilloscope plus the stray capacitance of the connection cable. Thanks to this method the frequency response of the circuit become flat and no overshoot or undershoot are observed.

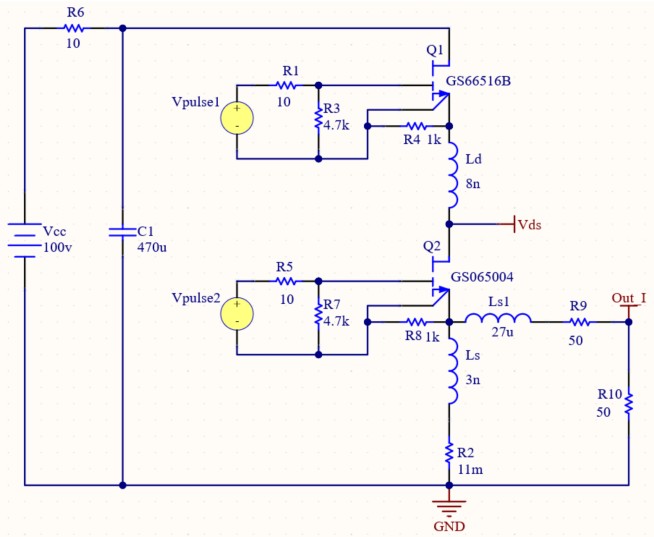

**Figure 5.** Simulated circuit with compensation network.

In this work we propose a similar method to compensate the effects of stray inductance. The value of the additional inductance can be calculated by analyzing the total impedance of the circuit. In particular, the time constant of the shunt (Ls/R2) must be equal to that of the branch Ls1, R9 and R10 resulting in

$$Ls1 = \frac{Ls}{R2}\,(R9 + R10) \tag{1}$$

This result is easily justified if we look at the two circuits of Figure 6. In both cases, the upper part of the circuit of Figure 3 is schematised as a current generator that injects the drain current into the measurement circuit. In Figure 6a the case of the resistive shunt is considered including the effects of the parasitic inductance Ls. In the circuit of Figure 6b, we have inserted the compensation inductance Ls1.

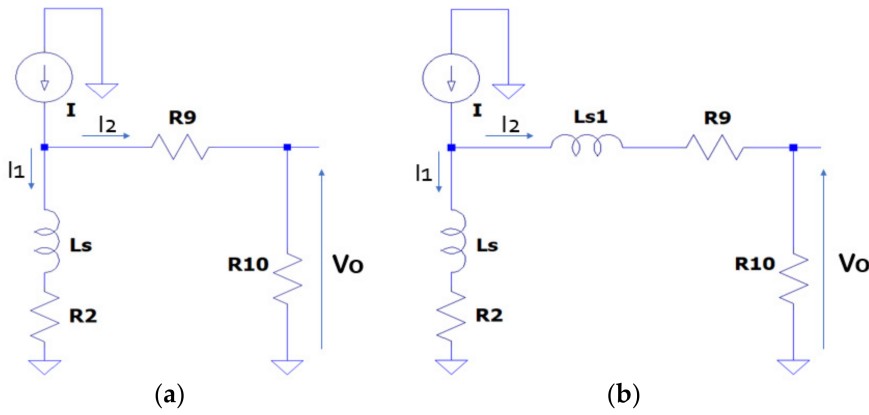

(**a**)                                        (**b**)

**Figure 6.** Schematic of measure circuit considering parasitic inductance Ls (**a**) and the compensation parameter Ls1 (**b**).

The output voltage Vo, calculated across the resistance R10 in the frequency domain results for the circuit (a) in

$$Vo(\omega) = \frac{R2 + j\omega Ls}{R9 + R10}\,R10\,I(\omega) \tag{2}$$

and it depends on both the value of the inductance Ls and the frequency.

For the circuit of Figure 6b, the output voltage is obtained as

$$Vo(\omega) = \frac{R2\left(1 + \frac{j\omega Ls}{R2}\right)}{(R9 + R10)\left[1 + \frac{j\omega(Ls+Ls1)}{R2+R9+R10}\right]}\,R10\,I(\omega) \tag{3}$$

By appropriately choosing the value of the inductance Ls1, it is possible to make the coefficients of the imaginary parts equal. In this case, Equation (3) is simplified and the relationship between voltage and current becomes purely resistive.

Considering the resistance and inductance values of the circuit of Figure 5 this corresponds to the insertion of an inductor Ls1 = 27.27 μH.

Figure 7 reports the result of the PSpice simulation in this last case. In the figure, the waveform in green, $I_{shcomp}$, represents the current measured using compensated shunt that is perfectly superimposable to the drain current of the DUT I(d). Moreover, we can observe that no additional time constant is introduced, as in the case of the standard RC network of oscilloscope probe.

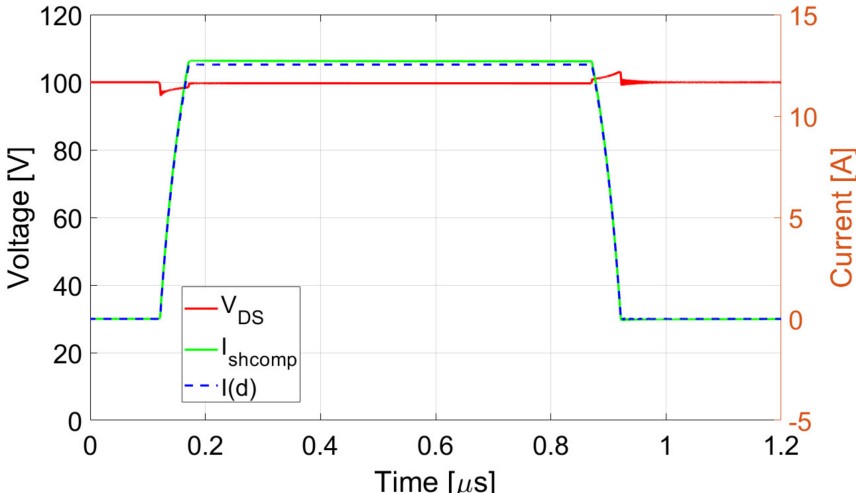

**Figure 7.** Simulated drain voltage (red) and drain current waveforms. $I_{shcomp}$ curve (green) represents the estimated current using compensation network, blue curve I(d) the actual current flowing into the device.

This solution, although simple, requires the insertion of a rather high compensation inductance in most practical cases not available. This problem can be overcome by using a two-steps compensation network such as that of Figure 8. In this case, the insertion of a low-value resistor (1 Ω) allows us to work with an inductance of hundreds of nH.

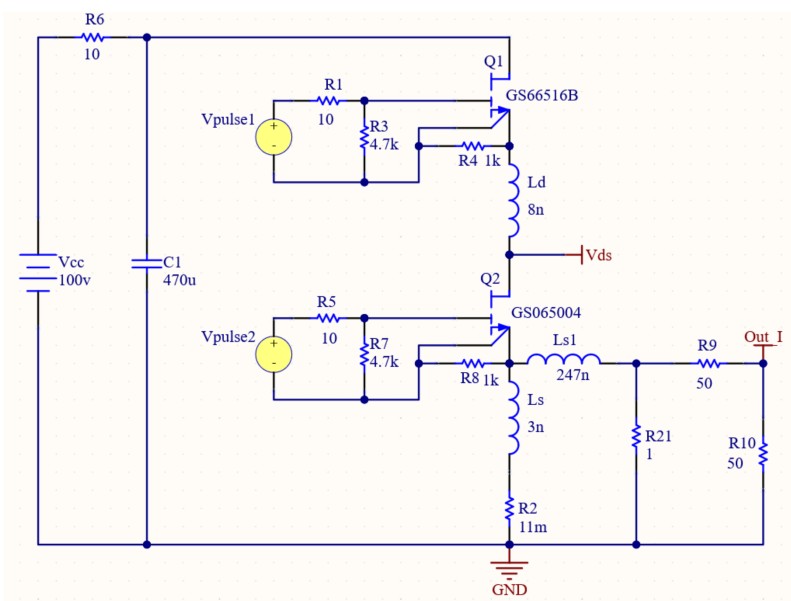

**Figure 8.** Simulated circuit with two steps compensation network.

The results on the measured current coincide with the one-step network and are not reported for the sake of brevity.

## 3. Experimental Tests

The circuit of Figure 1 was used to experimentally verify the proposed method. The tests were conducted using a 600 V-30 A Infineon GaN HEMT.

The test conditions were: $V_{DS} = 75$ V, $V_{GS} = 5$ V. The SC duration of 15 μs and the low drain voltage were chosen to avoid high thermal stress. This choice does not affect the results obtained as our goal is to demonstrate the feasibility of compensation. For the experimental tests, we used the two-step compensation network with the R and L values

(see Figure 8) coincident with those obtained from the LTSpice simulation. As can be seen from Figure 9, which shows a detail of the experimental circuit, to accurately set the inductance Ls1, which is 247 nH, we used a tunable inductor in the range 160–300 nH. The use of a variable inductor also allows us to compensate for any changes in the parameters related to the processes of realization of the devices and circuits that are not taken into account by the Spice models [14].

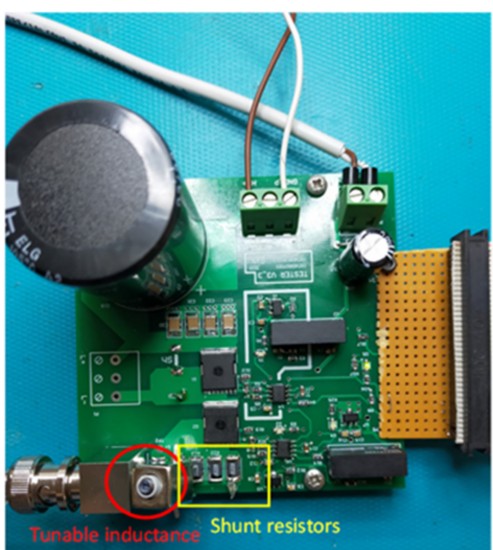

**Figure 9.** Detail of experimental circuit for SC tests with compensation network.

Finally, Figure 10 shows the $V_{DS}$ voltage and the current evaluated in the two cases. The $I_D$ current obtained by direct measurement of the voltage across the shunt resistors is shown in green. The initial value of the current is about 80 A and then starts to decrease to reach about 9 A after 15 μs. In blue we have the current measured using the proposed two-step compensation network. As can be seen, the measured current starts from at 17 A a considerably lower value than the previous one and then settles on 9 A.

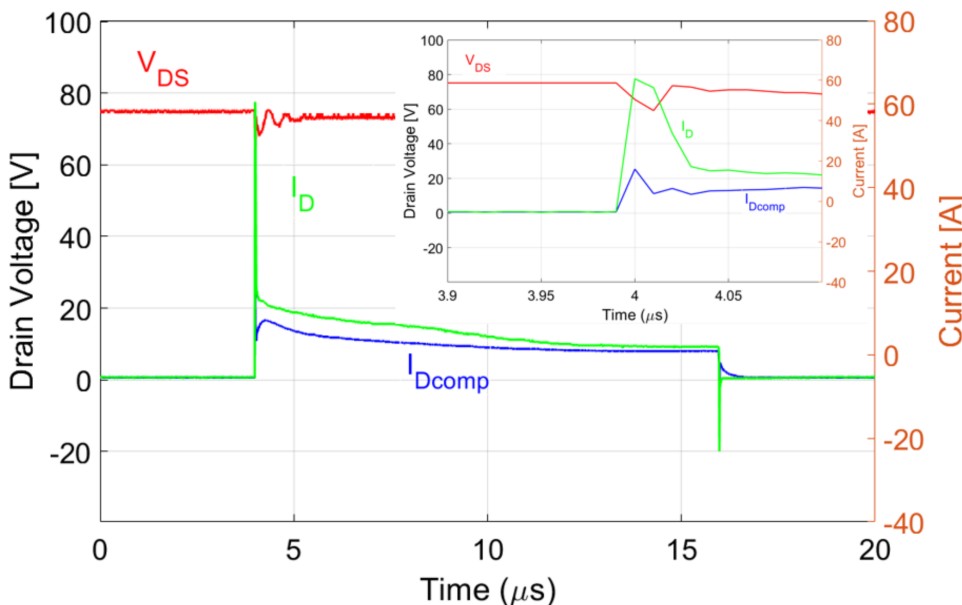

**Figure 10.** Experimental currents evaluated at $V_{DS}$ = 75 V and $V_{GS}$ = 5 V with (blue) and without (green) compensation network. The inset shows a zoom of the first microsecond of the transient.

## 4. Conclusions

A simple method was presented to obtain an accurate current measurement even in very fast switching cases using the resistive shunt technique. The idea is to use a compensation network to minimize the effects of parasitic inductances introduced by the resistors. This network, in the simplest case, coincides with the insertion of an inductor in series with the resistances of the measuring circuit. The proposed method was validated through a simple circuit model and Pspice simulations. The technique was finally used for the determination of the drain current of a GaN device during the short circuit. More generally, the method can be used in all applications involving the characterization of very fast devices.

**Author Contributions:** Conceptualization, C.A., F.V., G.B. and A.S.; methodology, C.A.; software, E.M., S.P.; validation, C.A., L.C., R.D.F. writing—original draft preparation, A.S.; writing—review and editing, A.S. All authors have read and agreed to the published version of the manuscript.

**Funding:** This research was funded by Ministry of University and Research as Research Projects of National Relevance n: 2017 WA5ZT3.

**Institutional Review Board Statement:** Not applicable.

**Informed Consent Statement:** Not applicable.

**Conflicts of Interest:** The authors declare no conflict of interest.

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
