# Peer review of "An Accurate Switching Current Measurement Based on Resistive Shunt Applied to Short Circuit GaN HEMT Characterization"

_applsci, doi:10.3390/app11199138_

Round 1
Reviewer 1 Report
The manuscript suggests a simple LR network for compensating transient overvoltage signals in a shunt resistor for current detection in a short-circuit test bench for fast GaN switching transistors. It is worth to highlight the issue with the parasitic inductances of shunt resistors and the presented compensation network is a feasable approach which is worth to publish. However, the overall increased time constant for current detection as consequence of the proposed network is not discussed. Further, the manuscript needs additional minor improvements.
Here are the details:
(1) Benefits of GaN HEMT devices for power conversion are presented in the introduction but when it comes to switching speed the specific benefits of GaN RF amplifying devices (usable up to hundreds of GHz) are quoted. This is not really consistent since device technology of power-electronic and RF GaN devices (although both are based on the GaN HEMT concept) is different. Better highlight the small device capacitances and small gate charge of GaN power-switching transistors when highlighting the superiour switching speed.
(2) In page 2, line 71, current peaks in the "first nanoseconds" are discussed. Please quote the related figure! (Figure 2, I guess)
(3) Figure 2 displayes drain currurent transients with µs (probably 100 ns) resolution but defenitely not resolution for the "first nanoseconds". Please correct.
(4) From the text in the last paragraph of page 2, I expect to find a clear distingtion between the "very high current peak in the first nanoseconds" and the "current decrease due to the occurance of thermal phenomena". But I am not able to see such separation in the current transients in Fig. 2. At exacly which time does the "very high current peak" ends?
The sitation is different for the current transients in Fig. 9. It is probably more wise to only discuss the transients of Fig. 9 and to skip Fig.2.
(5) A higher time resoltion of the current peak in Fig. 9 (as additional graph) would be helpful to be able to better judge on the matching of the simulated and experimentally obtained current transients.
(6) Please use the correct unit for time in Fig. 9.
(7) The legends, scales and units in Figs. 4 & 6 are by far too small to be readable. Please improve.
(8) The stray inductance of the shunt resistor was chosen as 3 nH for the simulations. Please provide arguments for choosing this value. (The authors report on a measured stray inductance of 3 nH for the power drain circuit but this is not identical to the shunt resistor.)
(9) The authors highlight the high bandwidth of current measurements via shunt resistors. However, displayed current transients with using the compensation network in both, simulation and experiment, show significantly increased time constants (~ > 200 ns) that are of similar magnitude as the total overvoltage spikes. It is very important to discuss the impact of the propesed network on the current detection time constant to judge of the practical usefullness of the propesed circuit.
Author Response
The manuscript suggests a simple LR network for compensating transient overvoltage signals in a shunt resistor for current detection in a short-circuit test bench for fast GaN switching transistors. It is worth to highlight the issue with the parasitic inductances of shunt resistors and the presented compensation network is a feasable approach which is worth to publish. However, the overall increased time constant for current detection as consequence of the proposed network is not discussed. Further, the manuscript needs additional minor improvements.
Thanks for the comments. The observations have all been acknowledged. The figures have all been improved and made clearer and more legible. English has been extensively revised. Below is a detailed description.
Here are the details:
- Benefits of GaN HEMT devices for power conversion are presented in the introduction but when it comes to switching speed the specific benefits of GaN RF amplifying devices (usable up to hundreds of GHz) are quoted. This is not really consistent since device technology of power-electronic and RF GaN devices (although both are based on the GaN HEMT concept) is different. Better highlight the small device capacitances and small gate charge of GaN power-switching transistors when highlighting the superiour switching speed.
As suggested, we have tried to better highlight the advantages of GAN technology for RF applications.
(2) In page 2, line 71, current peaks in the "first nanoseconds" are discussed. Please quote the related figure! (Figure 2, I guess)
(3) Figure 2 displayes drain currurent transients with µs (probably 100 ns) resolution but defenitely not resolution for the "first nanoseconds". Please correct.
(4) From the text in the last paragraph of page 2, I expect to find a clear distingtion between the "very high current peak in the first nanoseconds" and the "current decrease due to the occurance of thermal phenomena". But I am not able to see such separation in the current transients in Fig. 2. At exacly which time does the "very high current peak" ends?
The sitation is different for the current transients in Fig. 9. It is probably more wise to only discuss the transients of Fig. 9 and to skip Fig.2.
The purpose of figure 2 is to show the typical short circuit curves obtained with our experimental set-up. For this reason, the figure is still present, but we have tried to better explain the observations related to the measured drain current.
(5) A higher time resoltion of the current peak in Fig. 9 (as additional graph) would be helpful to be able to better judge on the matching of the simulated and experimentally obtained current transients.
(6) Please use the correct unit for time in Fig. 9.
Figure 9 has been improved. The time scale was correct. The comparison of the currents relates only to experimental measurements. In one case we evaluated the current using the classic resistive shunt, in the other by adding the compensation network.
(7) The legends, scales and units in Figs. 4 & 6 are by far too small to be readable. Please improve.
Figure 4 and 6 have been improved.
(8) The stray inductance of the shunt resistor was chosen as 3 nH for the simulations. Please provide arguments for choosing this value. (The authors report on a measured stray inductance of 3 nH for the power drain circuit but this is not identical to the shunt resistor.)
We have reported the exact values of the inductances and the measurement methods.
(9) The authors highlight the high bandwidth of current measurements via shunt resistors. However, displayed current transients with using the compensation network in both, simulation and experiment, show significantly increased time constants (~ > 200 ns) that are of similar magnitude as the total overvoltage spikes. It is very important to discuss the impact of the propesed network on the current detection time constant to judge of the practical usefullness of the propesed circuit.
Regarding this point we have added some considerations in the text and a reference.
Reviewer 2 Report
In this manuscript, the method to measure drain current during short-circuiting of fast devices such as GaN HEMTs has been proposed using compensation network. For more academic or technical contributions, the authors should make up for the followings.
(1) The abstract and introduction of this manuscript has been mainly descripted as a common content or problem that can occur in current measurements of high-speed switching devices. The core contents of the manuscript (details of the compensation network) should be properly included in the manuscript's abstract and introduction.
(2) The extraction method of the stray inductance derived from the shunt resistor and the determination method of the additional inductance of the compensation network shall be presented in detail.
(3) Because there are many grammatical errors in English sentences (for example, “The resistive shunt is by far the simplest sensor to make, cheap and easy to integrate, it also allows the measure of both DC and AC components at high frequency [6].”), the improvement of English sentences should be made.
Author Response
In this manuscript, the method to measure drain current during short-circuiting of fast devices such as GaN HEMTs has been proposed using compensation network. For more academic or technical contributions, the authors should make up for the followings.
Thanks for your review and helpful suggestions. All your comments have been welcomed. Below is the detail.
- The abstract and introduction of this manuscript has been mainly descripted as a common content or problem that can occur in current measurements of high-speed switching devices. The core contents of the manuscript (details of the compensation network) should be properly included in the manuscript's abstract and introduction.
We have improved the abstract and the introduction. Details of the compensation network have been added in the text.
- The extraction method of the stray inductance derived from the shunt resistor and the determination method of the additional inductance of the compensation network shall be presented in detail.
We have specified the values and the method of measurement of the inductances of the circuit and the shunt.
(3) Because there are many grammatical errors in English sentences (for example, “The resistive shunt is by far the simplest sensor to make, cheap and easy to integrate, it also allows the measure of both DC and AC components at high frequency [6].”), the improvement of English sentences should be made.
The work has undergone an extensive revision of English.

Round 2
Reviewer 2 Report
The authors have made great efforts to revise the manuscript, but there are some inadequate parts. Additional modifications must be made for publication.
(1) The abstract and introduction of this manuscript has been mainly descripted as a common content or problem that can occur in current measurements of high-speed switching devices. The core contents of the manuscript (details of the compensation network) should be properly included in the manuscript's abstract and introduction.
- In Abstract: “In this paper a simple passive compensation network is proposed, which allows an accurate measurement of the current using the resistive shunt even in the presence of very fast devices.” → “In this paper, a passive compensation network is proposed, which is formed by adding an inductor to the voltage measurement circuit and allows an accurate measurement of the current using the resistive shunt even in the presence of very fast devices.”
- In Introduction: “In this paper we show how it is possible to eliminate measurement artefacts through the insertion of a simple compensation network.” → “In this paper we show how it is possible to eliminate measurement artefacts through the insertion of a simple compensation network which is formed by adding an inductor to the voltage measurement circuit.”
(2) The extraction method of the stray inductance derived from the shunt resistor and the determination method of the additional inductance of the compensation network shall be presented in detail.
- The extraction method of the stray inductance was well organized.
- The determination method of the additional inductance (Ls1) of the compensation network has been developed mathematically, but Ls1 calculated by Ls/R2·(R9/R10) equation is not 25 uH. The method of determining Ls1 is very important in securing the legitimacy of this paper and therefore the authors must present the justification of Ls1 equation.
(3) Some sentence modifications are required.
- In line 63: “Figure 1. Test circuit used for the SC measurements.” → “Figure 1. Test circuit used for the short circuit measurements.”
- In line 100: “Using the simulator allows us to evaluate the effective current in the device and to highlight the effect of the parasitic parameters on the measurement.” → “The simulator helps us to evaluate the effective current in the device and to highlight the effect of the parasitic parameters on the measurement.”
- In line 103: “Fig. 3 shows the simulated circuit to reproduces the pulsed tests. The devices used are two 650 V Enhancement Mode GaN Transistor marketed by GaNSystem.” → “Fig. 3 shows the simulated circuit to reproduce the pulsed tests. The devices used are two 650 V enhancement-mode GaN transistor marketed by GaNSystem.”
- In line 114: “Figure 3. Simulated circuit.” → “Figure 3. Test circuit to observe the pulse characteristics.”
- In line 136: “In this case a capacitor in parallel to the voltage divider compensate the input capacitance of the scope plus the stray capacitance of the connection cable.” → “In this case a capacitor in parallel to the voltage divider compensates the input capacitance of the oscilloscope plus the stray capacitance of the connection cable.”
- In line 167: “equation (2): V0(w)=(R2+jwLs)·R10·I(w)/(R9+R10)” → “equation (2): V0(w)=(R2+jwLs)·R10·I(w)/(R9+R10+jwLs1)”
- In line 225: “The initial value of the current is about 80 A then starts to decrease to reach about 9 A after 15 us.” → “The initial value of the current is about 80 A and then starts to decrease to reach about 9 A after 15 us.”
Author Response
Thanks for the review.
All your comments have been taken into consideration. The text has been corrected. Regarding the determination of Ls1, the formula has been corrected and it has been specified in the text how it was used. In particular, the value of eq.1 is obtained by making simplifying hypotheses. This implies that the value of the inductance requires a tuning which in the experimental measure is obtained through the use of a variable inductance.
Round 3
Reviewer 2 Report
The authors have made great efforts to revise the manuscript, but some modifications and supplements are still needed.
(1) The abstract and introduction of this manuscript has been mainly descripted as a common content or problem that can occur in current measurements of high-speed switching devices. The core contents of the manuscript (details of the compensation network) should be properly included in the manuscript's abstract and introduction.
- Revision is OK.
(2) The extraction method of the stray inductance derived from the shunt resistor and the determination method of the additional inductance of the compensation network shall be presented in detail.
- Ls, R2, R9, and R10 are fixed values, but it is not understood that Ls1 has two values (by eq. (1) and by eq. derived from (3)). The authors should provide a valid explanation for this.
(3) Some sentence modifications are required.
- In line 55: “In this paper we show how it is possible to eliminate measurement artefacts through the insertion of a simple compensation network which is formed by adding an inductor to the voltage measurement circuit.” → “In this paper we show how it is possible to eliminate measurement artefacts through the insertion of a compensation network which is formed by adding an inductor to the voltage measurement circuit.”
- In line 160: “A quick estimate of the value of the inductance Ls1 can be made considering that Rs << R9 and R10 and assuming that Ls << Ls1.” → “A quick estimate of the value of the inductance Ls1 can be made considering that R2 ≪ R9 and R10 and assuming that Ls ≪ Ls1.”
- In line 163: “To obtain that the ratio between the measured voltage and the current I becomes purely resistive, it is necessary to tune the value obtained.” → “To obtain that the ratio between the measured voltage and the current becomes purely resistive, it is necessary to tune the value obtained from equation (1).”
Author Response
Please see the attachment. AS
